# Variability of Diurnal Sea Surface Temperature during Short Term and High SST Event in the Western Equatorial Pacific as Revealed by Satellite Data

**Anindya Wirasatriya** [1,2,*], **Kohtaro Hosoda** [3], **Joga Dharma Setiawan** [2,4] **and R. Dwi Susanto** [5,6]

1   Department of Oceanography, Faculty of Fisheries and Marine Sciences, Diponegoro University, Semarang, Jawa Tengah 50275, Indonesia

2   Center for Coastal Disaster Mitigation and Rehabilitation Studies, Diponegoro University, Semarang, Jawa Tengah 50275, Indonesia; joga.setiawan@ft.undip.ac.id

3   Weathernews Inc. Makuhari Techno Garden, Nakase 1-3 Mihama-ku, Chiba-shi 261-8501, Japan; hoso-k@wni.com

4   Department of Mechanical Engineering, Faculty of Engineering, Diponegoro University, Semarang, Jawa Tengah 50275, Indonesia

5   Department of Atmospheric and Oceanic Science, University of Maryland, Maryland, MD 20742, USA; dwisusa@umd.edu

6   Faculty of Earth Science and Technology, Bandung Institute of Technology, Bandung, Jawa Barat 40132, Indonesia

*   Correspondence: aninosi@yahoo.co.id; Tel.: +62-2-4747-4698

**Abstract:** Near-surface diurnal warming is an important process in the climate system, driving exchanges of water vapor and heat between the ocean and the atmosphere. The occurrence of the hot event (HE) is associated with the high diurnal sea surface temperature amplitude ($\delta$SST), which is defined as the difference between daily maximum and minimum sea surface temperature (SST). However, previous studies still show some inconsistency for the area of HE occurrence and high $\delta$SST. The present study produces global $\delta$SST data based on the SST, sea surface wind data derived from microwave radiometers, and solar radiation data obtained from visible/infrared radiometers. The value of $\delta$SSTs are estimated and validated over tropical oceans and then used for investigating HE in the western equatorial Pacific. A three-way error analysis was conducted using in situ mooring buoy arrays and geostationary SST measurements by the Himawari-8 and Geostationary Operational Environmental Satellite (GOES). The standard deviation error of daily and 10-day validation is around 0.3 °C and 0.14–0.19 °C, respectively. Our case study in the western Pacific (from 110°E to 150°W) shows that the area of HE occurrence coincided well with the area of high $\delta$SST. Climatological analysis shows that the collocated area between high occurrence rate of HE and high $\delta$SST, which coincides with the western Pacific warm pool region in all seasons. Thus, this study provides more persuasive evidence of the relation between HE occurrence and high $\delta$SST.

**Keywords:** diurnal SST amplitude; hot event; western equatorial Pacific; three-way error analysis; microwave radiometry

## 1. Introduction

Sea surface temperature (SST) has a typical daily cycle, called diurnal SST. The generation of diurnal SST is mainly caused by the changes in solar heating as a result of day and night differences. The amplitude/range of the diurnal SST variation ($\delta$SST) is enhanced up to more than 3 °C under the

calm and clear conditions [1–4]. This large δSST may lead to the appearances of unnatural patches or strakes in the daily SST map if satellite-derived SSTs are simply averaged without considering the diurnal variation [5,6].

Near-surface diurnal warming is an important process in the climate system since it drives exchanges of water vapor and heat between the ocean and the atmosphere [7–9]. Bernie et al. [10] and Li et al. [11] indicated that the diurnal SST variation influences the atmosphere over the western Pacific warm pool. Gas exchange, such as carbon dioxide flux at the sea surface [12], is also affected by diurnal SST cycle, and it has also been suggested that including the diurnal cycle in the calculations of climate model could improve representations of climate variability [7]. Moreover, the diurnal air–sea interaction may play an essential role in the longer time scale of climate variabilities such as Madden–Julian oscillation (MJO), and El Niño and Southern oscillation (ENSO) [13–17]. For the shorter scale, δSST variation determines the formation of the short period and high SST phenomena, called hot event (HE) [18–20].

Wirasatriya et al. [20] defined HE as the occurrence of SSTs higher than the space-time dependent threshold (about 30 °C), with the minimum area of $2 \times 10^6$ km$^2$ and lasting for a period longer than six days. Statistically, they indicated the role of HE distribution for the formation of the western Pacific warm pool. Higher occurrence rates of HE correspond to higher climatological SSTs. Thus, these short term and high SST phenomena may have climatic consequences if they accumulate and then affect the long-term mean SST pattern in the western Pacific warm pool.

Furthermore, Wirasatriya et al. [21] proposed the possible mechanism that explains the relationship between the western Pacific warm pool and the HE occurrences. They described that the frequent occurrence of HE could maintain the warm isothermal layer in the western Pacific warm pool through the heat exchange from the surface layer to the deeper layer. During the development stage of HE, the heat is accumulated in the surface layer as a consequence of high solar radiation and low wind speed. Strong westerly wind during the decay stage of HE generates convergent currents that transport the surface accumulated heat to the deeper layer. They demonstrated that the period with frequent HE occurrence could maintain the warm mixed layer of the western Pacific warm pool.

The mechanism of HE occurrence cannot be separated from the variability of δSST. HE only occurs under the condition of high solar radiation and low wind speed. This condition is associated with high δSST [18–20]. The relation between HE occurrence and high δSST was firstly examined by Qin et al. [18] for 31 HE cases spread throughout the whole equatorial region during the period from 1999 to 2009. They used estimated δSST data at 1 m depth produced by Kawai and Kawamura [22]. Their δSST data were constructed using a parametric model from solar radiation, wind speed, and precipitation data with 0.25° grid interval. Their root-mean-square error (RMSE) ranges from 0.2 °C to 0.3 °C. Qin et al. [18] has shown that the large δSST corresponds to the large amplitude of HEs. However, there were some area differences between large δSST and the large amplitude of HEs. By improving the threshold of Qin et al. [18], Wirasatriya et al. [20] managed to identify 71 HEs, even only in the western equatorial region in a shorter period, i.e., from 2003 to 2011. Nevertheless, the spatial analysis of δSST variation was absent in their study. Thus, the purpose of this study is to investigate the relation between HE occurrence and δSST variation.

To achieve the objective, we reconstructed a global diurnal SST warming based on diurnal SST range (δSST) defined as the daily maximum minus minimum SST, calculated from satellite-derived wind and solar radiation. The basis of the estimation of δSST is the optimally-interpolated (OI) SST data using the blended microwave and visible/infrared measurements from polar-orbiting satellites. Thus, there are two main works presented in this paper, i.e., the production of δSST data, and its application for investigating HE in the western equatorial Pacific. Section 2 provides the dataset and method to produce δSST data while its validation and its relation with HE distribution in the western Pacific warm pool are presented in Section 3 and concluded in Section 4.

## 2. Dataset and Methods

### 2.1. SST Data Production

2.1.1. Diurnal SST Range and Foundation SST Estimates from Polar-Orbiting Satellite Data

The quality of blended multi-satellites SST product is essential for investigating the relation between HE occurrence and δSST variation. The global observation of SST by passive microwave radiometers was begun by the Advanced Microwave Scanning Radiometer for Earth Observing System (AMSR-E) on the Aqua satellite, which was launched on 4 May 2002. While the AMSR-E operation was terminated on 4 October 2011, its successor AMSR2 was launched on the Global Change Observation Mission-Water (GCOM-W) on May 18 2012 to extend the global observations of the AMSR series (AMSR-E and AMSR2). On the basis of data from the Tropical Rainfall Mapping Mission (TRMM) Microwave Imager (TMI), it had been noted that cloud-free microwave SST observations are useful in capturing short-term phenomena in the ocean [23], the air-sea coupling of SST and sea surface wind (SSW) around SST fronts [24,25], and in providing superior SST coverage compared to infrared measurements [26]. However, the TMI observation area was limited to within the low- and mid-latitude oceans, and without a lower frequency channel, the TMI 10 GHz channel was relatively insensitive to SSTs in low temperature ranges [27,28]. Thus, the 6–7 GHz channels of the AMSR-E and AMSR2 systems are essential for obtaining global SST values with high accuracy.

In addition, WindSat on the Coriolis satellite, which was launched on January 6 2003, is another passive microwave radiometer, conducting global SST observations on the 6 GHz channel. While the swath widths of these microwave radiometers are as narrow as 1000–1500 km, the combination of measurements from two microwave radiometers renders it possible to obtain a daily SST composite with wide data coverage [6]. Among the daily composite calculation acquired in this way, the diurnal variation of SST is a key data point in producing the SST field without pseudo-signals. Such pseudo-signals arise because the local equator crossing times on the ascending node (LTAN) of the two instruments are different: the LTAN of the AMSR series is 13:30, close to time with daily maximum SST, whereas the daily-minimum SSTs are frequently observed at local time sunrise around 06:00, which is the local equator crossing times of the WindSat descending node. A method of obtaining gridded diurnal-free SST data is described in [29], in which the SST data at various observation times are diurnally corrected to daily-minimum SST using solar radiation and SSW data. This diurnal-free SST corresponds to the concept of foundation SST ($SST_{fnd}$), which is defined as the "water column temperature free from diurnal variation" [30]. Comparing diurnally varying SST with non-varying SST reveals average net-heat flux differences of up to 10 $W/m^2$, with seasonal and interannual variations also apparent [31]. Therefore, calculating the diurnal SST range, as presented in this study, is a critical aspect to produce a blended multi-satellites SST product.

The formula for calculating diurnal effects is based on the research by Kawai and Kawamura [5]. The method of calculating diurnal SST range from satellite observations was described in Hosoda [6], in which the diurnal correction method was developed to estimate daily maximum/minimum SST at 1 m depth ($SST_{max/min;1m}$) from satellite measurements at the various local times, as follows:

$$SST_{max/min;1m} = c_0 + c_1 SST_{1st} + c_2 \ln(SSW) + c_3 SR^2 + c_4 SR^2 \ln(SSW) \qquad (1)$$

where SR and SSW are the daily-means of solar radiation and SSW speed, respectively, and $SST_{1st}$ is the first-guess SST, which can be provided from daily gridded data or satellite remote sensing. Coefficients $c_0$, $c_1$, $c_2$, $c_3$, and $c_4$ vary with satellites and can be seen in Hosoda [6]. δSST is defined as the difference between daily $SST_{max}$ and $SST_{min}$. A number of formulations similar to this diurnal correction model have been proposed [32–35].

Based on this method, a global daily minimum SST or foundation SST was produced from OI multi-satellite measurements [29]. This dataset is primarily based on microwave SST observations from global sun-synchronous satellites: AMSR-E, WindSat, and AMSR2. The daily gridded data

are available from January 2003, with a spatial resolution of 0.1°. Additionally, the infrared SST measurements made by Moderate Resolution Imaging Spectroradiometers (MODIS) on the Terra and Aqua satellites were also merged to reproduce sub-mesoscale structures in the ocean in our SST product. The validation of this blended multi-satellites SST product against in situ data from the drifting buoys and Argo-floats shows that the root-mean-square error (RMSE) ranges from 0.46 °C to 0.48 °C [29].

In this study, Equation (1) was applied to estimate δSST with 0.1° spatial grid size from our blended multi-satellites SST product. The inputs are the daily means of solar radiation and SSW. SSW data are daily composites of microwave radiometer observations based on AMSR-E, AMSR2, WindSat, the Special Sensor Microwave Imager (SSM/I), and the Special Sensor Microwave Imager Sounder (SSMIS) series. These daily SSW composite data (0.25° × 0.25°) were adjusted with bicubic interpolation to obtain SSW values on the δSST calculation grid point (0.1°). For the solar radiation data, we primarily used the JAXA Satellite Monitoring for Environmental Studies (JASMES) dataset, which was produced from visible radiometer data from sun-synchronous satellites. The daily mean solar radiation data were prepared with a spatial grid size of 0.1° from original 0.05° grid data. An example of δSST as a function of daily mean solar radiation and SSW is shown in Figure 1, in which the first-guess SST is assumed to be 25 °C as $SST_{fnd}$. The relationships between the first-guess SST, SSW, and solar radiation were empirically derived by the co-location of satellite and drifting buoy measurements, as [6].

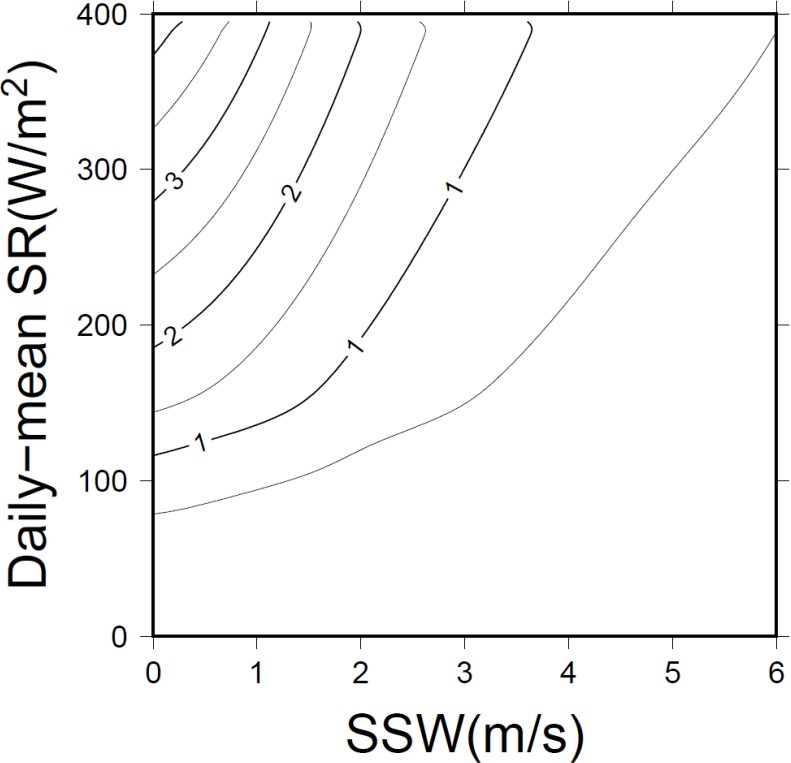

**Figure 1.** Diurnal sea surface temperature range (δSST) estimated from Equation (1) as a function of daily mean solar radiation (SR) and sea surface wind (SSW) used in this study. The first-guess sea surface temperature (SST) is assumed to be 25 °C as the foundation SST. The contour is δSST with the interval 0.5 °C.

2.1.2. Intercomparison Data from in situ and Geostationary Satellite Observations

In Situ SST Data

This study used in situ SST measurements based on the tropical moored buoy networks [36], consisting of the Tropical Atmosphere Ocean and Triangle Trans Ocean buoy Network moorings (TAO/TRITON) array, Research Moored Array for African–Asian–Australian Monsoon Analysis and

Prediction (RAMA), and Prediction and Research Moored Array in the Atlantic (PIRATA). The locations of the moored buoy networks used in this study are shown in Figure 2 alongside the validation results. Based on high temporal resolution (≤ 1 h) of each buoy, the δSST was calculated as daily maximum minus minimum. The daily minimum SST was determined near to local sunrise (LST 6:00 ± 2 h), and maximum SST was within local afternoon (LST 12:00–16:00).

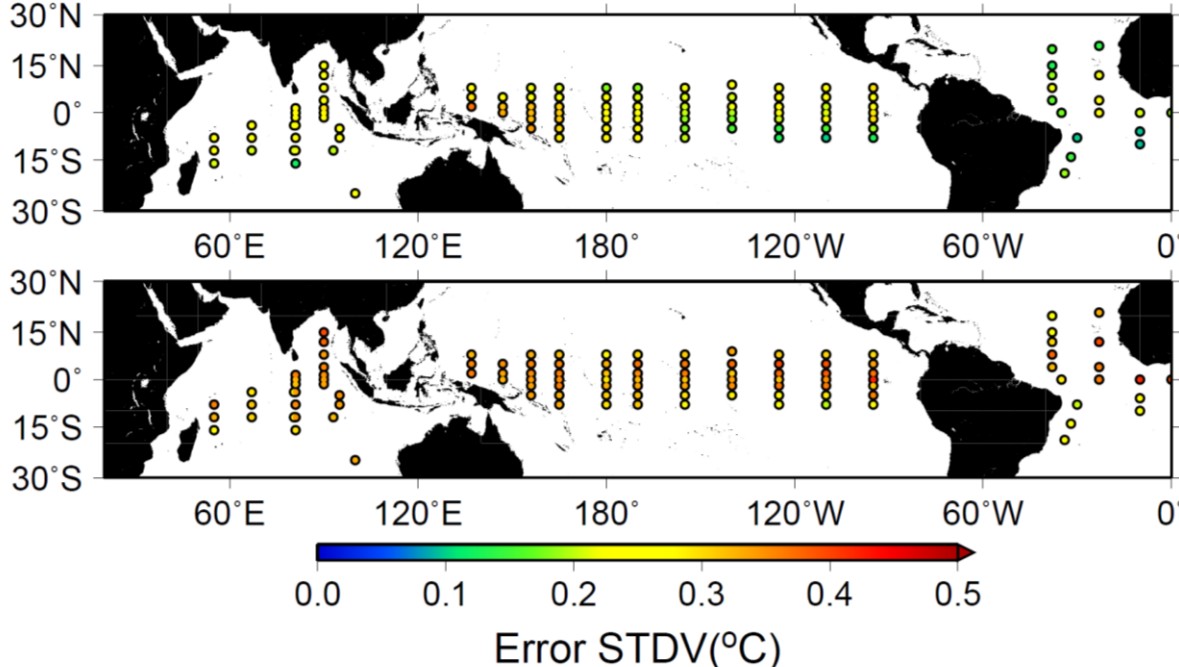

**Figure 2.** Standard deviation distributions for the daily satellite-based SST estimates against each moored buoy measurement. (Top) diurnal SST range estimates, and (bottom) foundation SST estimations.

Geostationary Satellite-Based SST Data

Infrared radiometers aboard geostationary meteorological satellites can also provide high-resolution time series of SST (≤1 h) in low and mid-latitude areas if the pixels are under clear sky condition. In this study, we used two satellite products that cover the tropical Pacific i.e., Geostationary Operational Environmental Satellite (GOES) and Himawari-8. The GOES series Level3 6 km resolution SST data [37] are used for the eastern Pacific and Atlantic oceans (combined from GOES East and GOES West) with a temporal resolution of 1 h. GOES SST data have been provided alongside cloud screening information using the Bayesian approach since 2008 [38]. For this study, the threshold cloud contamination possibility is set to 2% to obtain accurate SST with a wide coverage. The spatial and temporal coverages of the GOES SST are 45°S–60°N, 180°–30°W, and January 2008–December 2015. The hourly SST estimates derived from the Himawari-8 satellite with a spatial resolution of 2 km have been provided by the JAXA [39]. The spatial and temporal coverages used of these data in this study are 60°S–60°N, 80°E–160°W, and August–December 2015. Only those SST data flagged as the "best quality" level were used in the comparison of the present study. The definitions of daily minimum and maximum SST are the same as for the in situ observations. Both geostationary SST datasets were re-gridded to 0.1° × 0.1° and compared with our product.

## 2.2. Datasets for HE Analysis

Daily New Generation Sea Surface Temperature for Open Ocean (NGSST-O-Global-V2.0a) was used for the HE identification. This dataset is produced by merging SST observations acquired by two satellite microwave sensors (AMSR-E onboard Aqua and WindSat onboard Coriolis) with a grid interval of 0.25°. An optimal interpolation technique was applied for merging, using decorrelation

scales derived by Hosoda [40] after applying diurnal correction described in Hosoda [6]. The RMSE of this dataset is 0.43 °C. We used six-hourly reanalysis data from the Japanese 25-year Reanalysis (JRA-25)/Japan Meteorological Agency (JMA) Climate Data Assimilation System (JCDAS) on a 1.25° horizontal grid for wind speed [41] and daily net surface solar radiation on a 1° × 1° grid for 2003–2009 from the International Satellite Cloud Climatology Project (ISCCP) dataset [42]. The grid intervals of these datasets were interpolated into 0.25° to match with NGSST-O data.

For investigating the climatology of δSST of HE in the western equatorial Pacific, we compared the composite of δSST of 71 HEs identified by Wirasatriya et al. [20] during 2003–2011 with the occurrence frequency of the identified HEs shown in their research. We also compared the relative frequencies of δSST from 2003 to 2011 inside the area of HE occurrence frequency more than 5% and inside the area of individual HE during the HE period, development stage, and decay stage. The definitions of HE period, development stage, and decay stage are described in Wirasatriya et al. [43].

## 3. Results

### 3.1. SST Validation

Figure 3 presents frequency diagrams comparing buoy and satellite δSST and SST$_{fnd}$ in the tropical oceans. In the first approach using the traditional method of validation, in situ moored buoy observations are considered as truth data. The calculation was conducted using all available data from 2003 to 2015. In Figure 3, the upper panels show comparisons between daily data, while the 10-day mean data comparisons are given in the lower panels. The match-up-data density was calculated as the percentage of match-ups in a 0.1 °C × 0.1 °C grid in comparison with the total number of match-ups. If the match-up-data density value is less than 0.01%, the box is colored white. Absolute values of biases by both δSST and SST$_{fnd}$ were less than 0.05 °C, while their standard deviation (STDV) was 0.25 °C and 0.41 °C, respectively. The STDV of δSST is smaller, but the match-up data in Figure 3a have a wide distribution. The geographical distributions of the STDV are shown in Figure 2. Large STDVs of δSST (>0.3 °C) were located within the western tropical Pacific (140–160°E). However, the uneven distribution of such large errors was not seen in the SST$_{fnd}$ estimations. This wide distribution is probably due to the traditional approach of validation. We improve the approach by using a three-way error analysis described in the next paragraph. In contrast, in the 10-day mean comparison, this wide distribution was reduced, and the STDV of δSST was 0.12 °C. This STDV reduction was also found in the 10-day mean comparison of SST$_{fnd}$. This means that, while there is room for improvement in the reproduction of short-term variability of both SST$_{fnd}$ and δSST, the long-term analysis using several days or monthly mean is suitable for use in climate analysis.

The traditional method of satellite SST validation against the in situ SST as shown in Figures 2 and 3 left some problems. The in situ SST, which is assumed as the true SST, may not be consistent in terms of the depth of measurement depending on the buoy types. Lack of instrument maintenance, especially for drifting buoy, may affect the accuracy of the observed SSTs [44]. Furthermore, potential errors of traditional validation also can emerge due to the uncertainty differences between skin SSTs obtained from satellite measurements and bulk SSTs measured from in situ measurements. To overcome those problems, O'Carroll et al. [45] developed a three-way error analysis that considers these differences and corrects them where possible. The three-way error analysis, or triple collocation, is employed to estimate the unknown errors of three independent measurements, without assuming that any one system is able to observe the truth data perfectly. The concept of the three-way error analysis by O'Carroll et al. [45] is as follows:

If the errors in the three independent observation systems (i; j; k = 1; 2; and 3) are uncorrelated, then the variance of errors in each observation type $\sigma^2_i$ are expressed as,

$$s_i^2 = \frac{1}{2}\left(V_{ij} + V_{ki} - V_{jk}\right) \tag{2}$$

where $V_{ij}$ gives the variance of the difference between two observation types i and j. This relationship has been recently applied to SST validation analyses [46–48], and it is used to inter-compare daily and 10-day mean SST ($SST_{fnd}$ and $\delta SST$) of the present study, with in situ measurements, and geostationary high-resolution datasets.

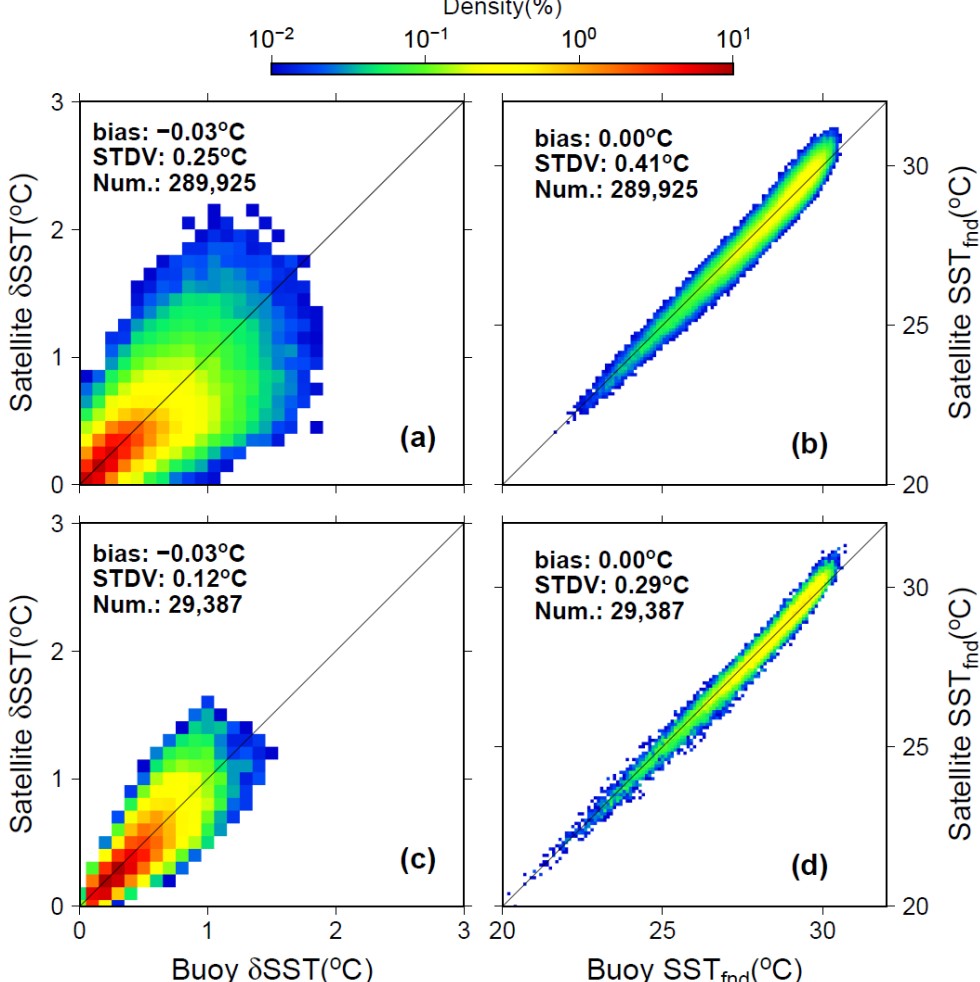

**Figure 3.** Frequency diagrams between satellite-based estimation and buoy observations of diurnal SST range ($\delta SST$, left column) and foundation SST ($SST_{fnd}$, right column). Daily step data comparisons ((**a**) and (**b**)), and 10-day averaged data comparisons ((**c**) and (**d**)). The density was calculated by the number of data values in each 0.1 °C × 0.1 °C grid. The comparison was conducted for all data from 2003 to 2015.

Table 1 shows that the in situ buoy measurements provided data with the lowest amount of errors as expected. The large errors of 0.55–0.64 °C in the geostationary SST may be partly due to cloud or aerosol contamination in the infrared algorithm, or observation depth differences since the skin SST is measured by infrared sensors, while the bulk temperature is given by the in situ sensors [41]. The errors in the infrared sensors were reduced to 0.20–0.39 °C by calculating the 10-day mean. The errors of $SST_{fnd}$ and $\delta SST$ in the products of this study were 0.27–0.47 °C and 0.14–0.23 °C for the daily and the 10-day comparisons, respectively. These are less than the geostationary measurements, and comparable to the in situ data. This result suggests that the blended microwave and infrared products are able to provide diurnal SST cycles ($\delta SST$) and $SST_{fnd}$ with higher accuracy than the infrared geostationary observations.

**Table 1.** Three-way error analysis derived individual satellite and buoy standard deviations (STDV) for diurnal range (δSST) and foundation SST (SST$_{fnd}$). Italic number shows the collocated match-ups in each condition.

| | This Study (Blended Product) | Geostationary | Buoy | Number | This Study (Blended Product) | Geostationary | Buoy | Number |
|---|---|---|---|---|---|---|---|---|
| | δSST (daily) | | | | δSST (10-day) | | | |
| STDV (GOES) | 0.27 °C | 0.64 °C | 0.24 °C | *3317* | 0.14 °C | 0.39 °C | 0.02 °C | *528* |
| STDV (Himawari-8) | 0.35 °C | 0.58 °C | 0.36 °C | *610* | 0.19 °C | 0.31 °C | 0.06 °C | *86* |
| | SST$_{fnd}$ (daily) | | | | SST$_{fnd}$ (10-day) | | | |
| STDV (GOES) | 0.47 °C | 0.55 °C | 0.42 °C | *2040* | 0.23 °C | 0.20 °C | 0.21 °C | *629* |
| STDV (Himawari-8) | 0.39 °C | 0.59 °C | 0.34 °C | *423* | 0.15 °C | 0.35 °C | 0.11 °C | *92* |

It should be noted that the data coverage of the geostationary sensors is strongly affected by cloud presence, even if the temporal sampling rates are ultra-high frequency (less than one hour). The medians of data coverages of the Himawari-8 δSST and SST$_{fnd}$ measurements were 46% and 57%, respectively, while those of the GOES were 11% and 26%, respectively. The smaller δSST coverage was due to the requirement of a persistent clear-sky condition throughout sunrise and afternoon, because δSST was calculated as the temperature difference between the two four-hour composites in these periods. The slightly larger coverage of the Himawari-8 data is likely due to the higher sampling rate by the Himawari-8, which has an original observation frequency of 10 min.

The temperature dependencies of the measurement errors derived from the three-way error analysis of the daily comparisons are presented in Figure 4. In the SST$_{fnd}$ estimation, no significant temperature dependency was identified. In contrast, a monotonic increase in error variance in δSST estimates made by this study was apparent at a temperature of ≥28 °C. This characteristic was not found in error variances by either geostationary or in situ measurements. This monotonic increase in error variance in δSST corresponds to the geographical distribution of STDV (Figure 2) mainly in the western tropical Pacific, which is characterized by warm water >28.5 °C and known as the Pacific warm pool [49].

The area of the present HE study is located in the western equatorial Pacific, which shows a large error of δSST data. Since HEs are categorized as short scale phenomena and δSST data have more accuracy for long term mean analysis, we need to re-validate the δSST data with TAO/TRITON buoys in the western equatorial Pacific for HE analysis. We compared the accuracy of δSST data between HE period and non-HE period to ensure the reliability of δSST data for HE study. The result shows that the bias and error STDV of δSST data against TAO/TRITON buoys in the western equatorial Pacific for 2003–2011 is −0.002 °C and 0.315 °C, respectively. For the non-HE period, the error STDV slightly decreases to 0.302 °C. In contrast, during the HE period, the STDV increases to 0.359 °C, and the bias turns into positive. This positive bias means that for HE period, the δSST data are mostly higher than δSST calculated from buoys. This condition may be caused by the extreme condition of low wind speed and high solar radiation that occurred during HE period. Thus, we suggest that the linear parameterization used for constructing δSST data should be evaluated especially for the extremely low wind speed and high solar radiation that co-occur. However, this dataset is reliable enough for the present study since the variation of δSST investigated in this study is much higher than its error.

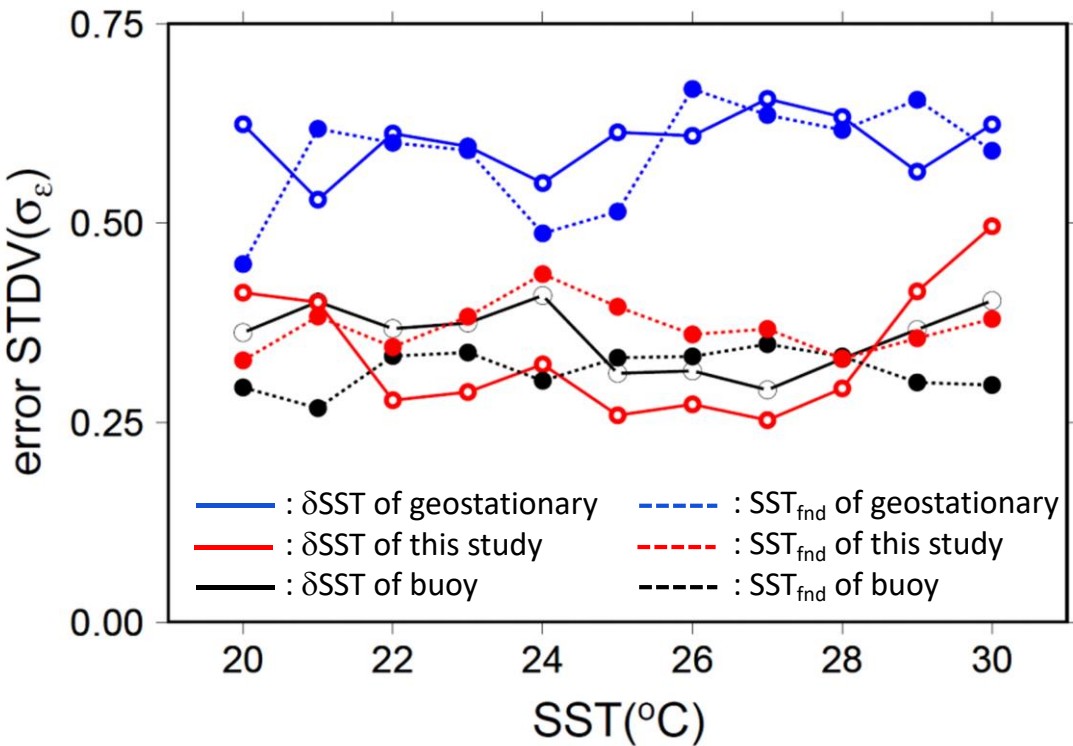

**Figure 4.** Standard deviation of error for each observation type as a function of SST. The calculation was carried out for each 1° grid using Geostationary Operational Environmental Satellite (GOES) and Himawari-8 data for the period 2008–2015, and August–December, 2015, respectively.

*3.2. Relation between δSST Variability and HE in the Western Equatorial Pacific*

To investigate the relation between δSST variability and HE, first we examine HE started on 16 December 2004 (hereafter HE041216) presented in Wirasatriya et al. [20] as a representative of HE in the western equatorial Pacific. Figure 5 shows the average map of δSST, solar radiation, and wind speed during the period HE041216 overlaid with the area of HE041216. The area of HE refers to the area with the SST more than the time-space dependent threshold (~30 °C) and lasting during the period of HE. The area of HE041216 agreed with the area of δSST more than 0.5 °C. The area of δSST more than 0.5 °C was consistent with the area of wind speed of less than 3 m/s and located at the area of solar radiation more than 200 W/m$^2$. Thus, this result supports the role of wind speed as the key factor for the HE occurrence as stated in Wirasatriya et al. [20].

For the climatological analysis, we show the distribution of δSST during the HE period for 2003–2011 in the western equatorial Pacific (Figure 6). Figure 6a shows that high δSST of more than 0.4 °C is distributed from 10°S to 10°N along the northern coast of New Guinea Island until 170°W. The high δSST distribution is collocated with the area of HE frequency occurrence of more than 5%. The seasonal change also shows the same tendency (Figure 6b,c). The northward (southward) shift of the area of high δSST distribution is followed by the northward (southward) shift of the area of high HE frequency occurrence during boreal summer (winter). This indicates the strong relation between HE and δSST distribution.

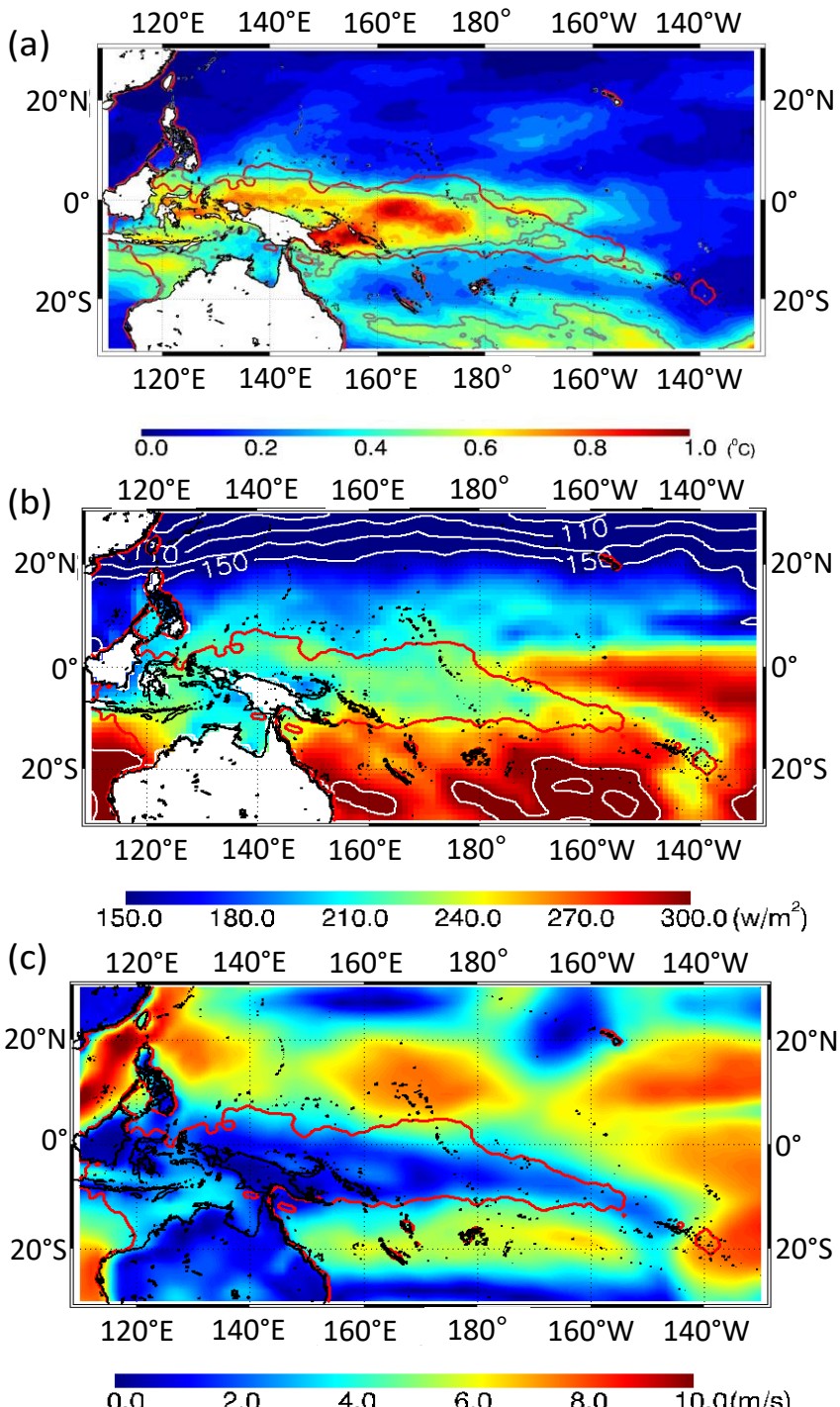

**Figure 5.** (**a**) δSST map, (**b**) solar radiation map and (**c**) wind speed map of HE041216. The red contour denotes the area of HE041216. The gray contour in (**a**) denotes the δSST of 0.5 °C. The white contour in (**b**) denotes solar radiation of less than 150 W/m² and more than 300 W/m².

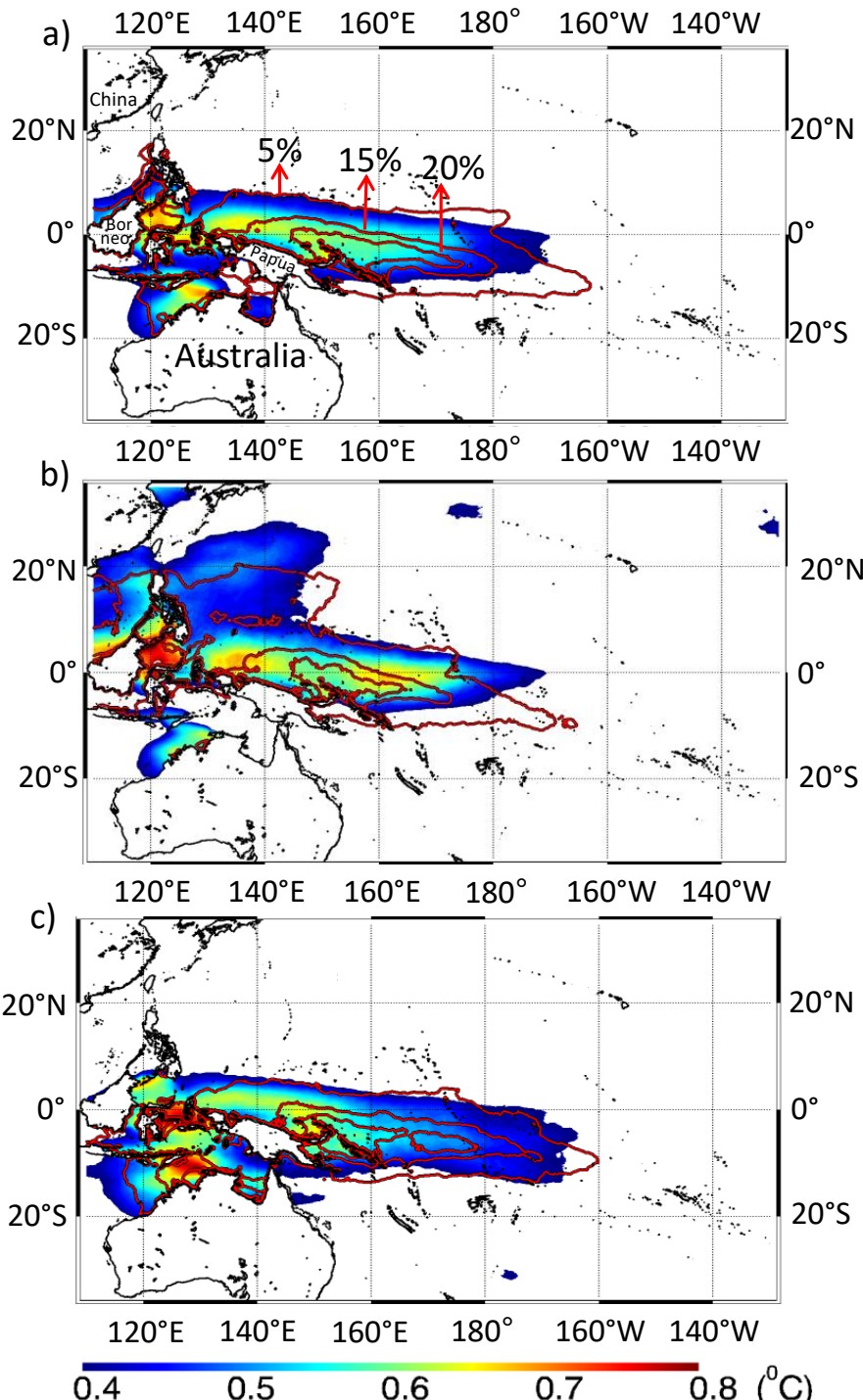

**Figure 6.** The composite of δSST map during hot event (HE) period overlaid with the contour of occurrence rates of HEs, shown in frequency per grid (%) during 2003–2011, for (**a**) the whole period (100% is 3163 days; ≈9 year period), (**b**) boreal summer (April–September; 100% is 1615 days), and (**c**) boreal winter (October–March; 100% is 1548 days).

For investigating the δSST variation in the development and decay stage of HE, the relative frequency of each value of δSST inside HEs during the development and decay stages of HEs is presented in Figure 7. The δSST inside HEs during the development stage is higher than the decay stage, indicated by the higher relative frequency of δSST of more than 0.4 °C. This result is consistent with Wirasatriya et al. [44], who showed the higher (lower) solar radiation (wind speed) during the

development stage than the decay stage. Furthermore, this study shows the relative frequency of δSST of more than 0.3 °C is higher inside the HE area during the HE period than outside the HE area during the HE period. This indicates the high δSST often occurs in the western equatorial Pacific, which makes the western equatorial Pacific favorable for HE generation.

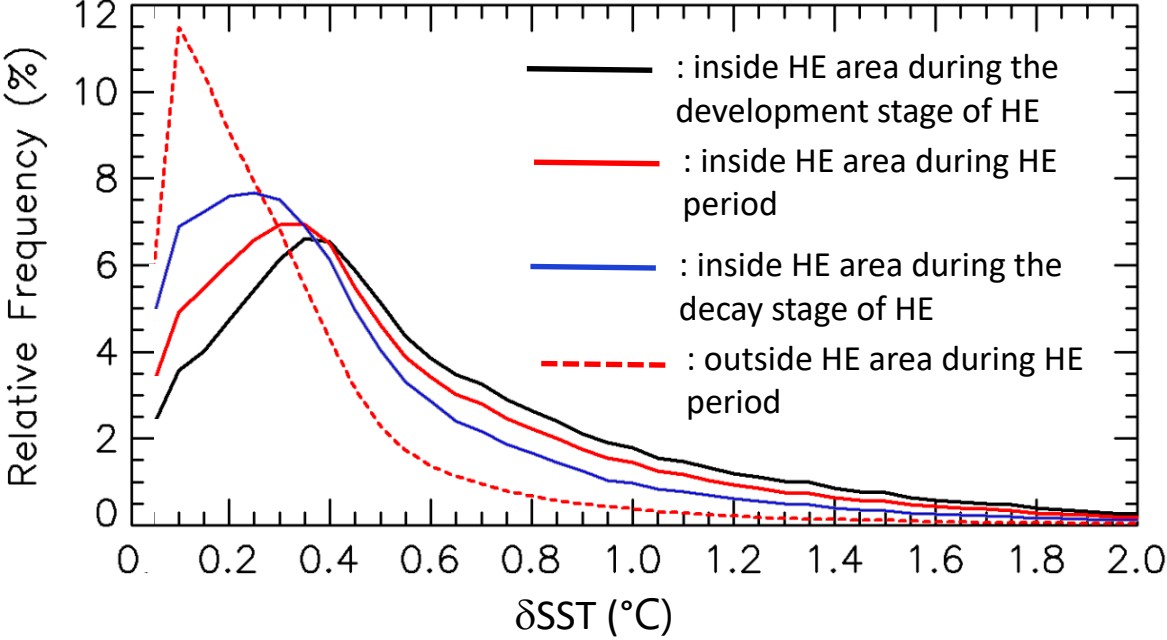

**Figure 7.** Histograms of the mean relative frequencies of δSST values, with discrete intervals of 0.05 °C.

## 4. Discussion

This study presents the production of δSST data based on the satellite-derived SST, SSW, and solar radiation. This dataset was produced based on Kawai and Kawamura [6] with the enhancement in the validation method using three-way error analysis, which can reduce the uncertainties between bulk and skin SST measurements [45]. Our product becomes the first δSST dataset that applies three-way error analysis, resulting in the significant improvement compared to other products. It is noted that the Equation (1) used for generating this product only applies to the open ocean. In the coastal area, the variable that influences the diurnal range of SST becomes more complex. For example, Wang et al. [50] demonstrated that tidal level and air temperature are responsible for the great diurnal SST variation in the coastal area. Furthermore, Maneghesso et al. [51] has reported the systematic positive bias of level four SST products against the in situ SST in the coastal upwelling area. Therefore, further improvement should be conducted to estimate the δSST for the coastal area. This task is left for future studies.

For the HE study, Wirasatriya et al. [20] has shown that the shifting pattern of δSST distribution is the result of the distribution of solar radiation and wind speed. Comparing the relation between the occurrence of HE and the occurrence of low wind speed and high solar radiation in Wirasatriya et al. [20] and the relation between the occurrence of HE and the occurrence of high δSST in Figure 6b,c, δSST distribution during the HE period shows a better relation with the HE occurrence rate than SR or wind speed distribution for both boreal summer and winter. Although Wirasatriya et al. [20] found that the low wind speed distribution became a key factor in the occurrence of HEs in the western equatorial Pacific, the area with a low wind speed of less than 4 m/s does not always coincide with the area with high occurrence rate of HE of more than 5%. This relation is because low wind speed should co-occur with high solar radiation to produce HE occurrence. Thus, neither only low wind speed nor only high solar radiation can be used as an indicator of HE occurrence. In the present study, we show that high

δSST can be a good indicator of HE occurrence in the western equatorial Pacific since both wind speed and solar radiation have been included for the calculation of δSST as described in Equation (1).

The climatological analysis of δSST during HE period in the equatorial region has also been shown by Qin et al. [18]. However, the inconsistency areas of HE and high δSST still appeared in their study. The area with high intensity of HE is located along the northern coast of Papua, while the area with high δSST is located along the equatorial line. The present study shows better consistency as shown in Figure 6. The improved threshold used in the present study i.e., SST threshold that excludes the seasonal variation, smaller areal size threshold, and shorter period threshold, resulted in the increased number of HE in a smaller study area. The increased number of HE may contribute to constructing the better composite of δSST of all HEs. Another difference is related to the relative frequency distribution of δSST. Qin et al. [18] showed that the relative frequency distribution of δSST follows the exponential function while in the present study it is positively skewed (Figure 7). This finding indicates the warmer SSTs in the western equatorial Pacific may promote the more frequent occurrence of high δSST than other areas in the equatorial region. However, the tendency of the relative frequency distribution of δSST is similar for both studies observing inside and outside the HE area during the HE period.

## 5. Conclusions

This paper describes the calculation, validation, and a climate study application of the diurnal SST range estimations using satellite observation data (SST, SSW, and SR). The validation was conducted using data from moored buoy arrays in tropical oceans: TAO/TRITON, PIRATA, and RAMA. The standard deviations of the estimations of in situ and satellite-based δSST are around 0.25 °C and 0.15 °C for daily and 10-day mean comparisons, respectively. In order to investigate the characteristic errors, a three-way error analysis was employed between satellite-based estimates, in situ observations, and geostationary measurements. The in situ measurements give the smallest errors while the geostationary measurements have the largest errors, and the errors of the satellite-based δSST lie in the middle and close to the errors of the in situ measurements. This result suggests that the measurements of the full diurnal cycle by a geostationary satellite equipped with ultra-high-resolution sensors, such as the 10-min resolution of Himawari-8, are compromised due to cloud cover. The blended microwave and infrared products are the essential basis for these diurnal SST and HE studies.

The application of δSST data for investigating HE in the western equatorial Pacific demonstrated a robust relationship between the occurrence of HE and high δSST, which is summarized as follows:

(a)  In the case study, the area of HE041216 occurrence coincided well with the area of δSST of more than 0.5 °C.

(b)  The climatological mean of δSST shows that high δSST of more than 0.4 °C is distributed from 10°S to 10°N along the northern coast of New Guinea Island until 170°W. The high δSST distribution is collocated with the area of HE frequency occurrence of more than 5%.

(c)  During boreal summer (winter) the high δSST distribution shifts northward (southward).

(d)  The δSST inside HEs during the development stage is higher than the one during the decay stage.

(e)  High δSST can be a good indicator of HE occurrence in the western equatorial Pacific since both wind speed and solar radiation have been included for the calculation of δSST.

**Author Contributions:** Conceptualization, A.W. and K.H.; methodology, A.W. and K.H.; software, A.W.; validation, K.H.; formal analysis, A.W. and R.D.S.; investigation, A.W., K.H.; resources, K.H., J.D.S.; data curation, K.H.; writing—original draft preparation, A.W. and K.H.; writing—review and editing, J.D.S.; visualization, A.W. and K.H.; supervision, R.D.S.; project administration, J.D.S.; funding acquisition, J.D.S. All authors have read and agreed to the published version of the manuscript.

**Funding:** This research is partially funded by non-APBN LPPM 2020, Universitas Diponegoro Contract 424 No: 329-121/UN7.6.1/PP/2020 and the Indonesian Ministry of Research and Technology/National Agency for Research and Innovation, and Indonesian Ministry of Education and Culture, under World Class University Program managed by Institut Teknologi Bandung. Anindya Wirasatriya thanks the World Class Profesor program managed by Indonesian Ministry of Education and Culture contract no. 101.2/E4.3/KU/2020. K. Hosoda was funded by the

Japan Aerospace Exploration Agency (JAXA) under the Global Change Observation Mission-Water (GCOM-W) 5th Research Announcement (JX-PSPC-434772). R. Dwi Susanto is supported by NASA grants#80NSSC18K0777 and NNX17AE79A through the University of Maryland, College Park.

**Acknowledgments:** The JASMES, AMSR-E, and AMSR2 datasets are available from the G-Portal of the Earth Observation Research Center, JAXA (https://www.gportal.jaxa.jp/gp/top.html). Himawari-8 SST data are available from the JAXA Himawari Monitor (http://www.eorc.jaxa.jp/ptree/). WindSat, SSM/I, and SSMIS data were downloaded from the Remote Sensing Systems website (http://www.remss.com/). The GOES SST data were provided by the National Oceanic and Atmospheric Administration (NOAA)'s Satellite and Information Service (NOAA/NESDIS: http://dx.doi.org/10.5067/GOES3-24HOR). The moored buoy data of the TAO/TRITON, PIRATA, and RAMA are distributed by the Pacific Marine Environmental Laboratory, NOAA (http://www.pmel.noaa.gov/tao/datadeliv/frames/main.html). NGSST-O-Global-V2.0a and $\delta$SST data are the property of Center for Atmospheric and Oceanic Studies, Tohoku University, Japan. Contact for data inquiry and request is Kohtaro Hosoda (khosoda@gmail.com). JRA-25 is provided by the Japan Meteorological Agency. This dataset can be downloaded at http://rda.ucar.edu/datasets/ds628.0/index.html#!access. ISCCP-FD data are from the courtesy of the NASA Goddard Institute for Space Studies and can be accessed at http://oaflux.whoi.edu.

**Conflicts of Interest:** The authors declare no conflict of interest. The funders had no role in the design of the study; in the collection, analyses, or interpretation of data; in the writing of the manuscript, or in the decision to publish the results.

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
