# Peer review of "Variability of Diurnal Sea Surface Temperature during Short Term and High SST Event in the Western Equatorial Pacific as Revealed by Satellite Data"

_remotesensing, doi:10.3390/rs12193230_

Round 1
Reviewer 1 Report
This paper investigates the relationship between Hot Events occurrence and diurnal SST amplitude (dSST) variation in the Western Equatorial Pacific by using satellite data. In order to do this the authors also present the methodology used to produce dSST data from satellite sensors and their validation. The research is interesting and well founded, but the article suffers from a lack of clarity, especially in the presentation of the results and their discussion.
It is obvious from reading the work that the authors start from a significant volume of information and data and that they have based their results on solid previous works. However, it is sometimes difficult to distinguish between the original results presented in this paper and previously published results, in part this is because the reader is frequently referred to figures from other papers (references 18 and 20) that are not shown in the document.
We recommend a REWRITE of the text taking into account the following recommendations:
- The objectives of the paper should be more clearly and extensively explained in the Introduction. It would be useful to explain whether the period of study is (2003-2011) or when and why the authors extend the period (1991-2002)
- In section 2 (Dataset and Methods) we suggest splitting the subsection “2.1.2. Intercomparison data from in situ and geostationary satellite observations” into two subsections, one to describe better the need for using in situ data and their characteristics or limitations, and the other to describe the satellite observations.
- We recommend dividing Section 3 Results and Discussion into two sections (3. Results and 4. Discussion). This would make clearer for the reader which are the new results presented by the authors and then separately compare them with other or previous works.
- The authors should not be making references to figures from other papers which are not presented in the text. This is done repeatedly: Lines 198-204, Line 308, Lines 331-332, Lines 361.This is very confusing, especially when they make statements such as: Line 309-310 “the relation between dSST and HE area is more consistent than the result shown by Qin et al. [18] in Fig. 4, i.e. the area of dSST more than 0.5°C was wider than the HE area”
- In Figure 5, It would be useful to explain how the authors have depicted the area of HE (red line). It seems that the method is explained in a previous study, Wirasatriya et al., but if the Figure shows your results in this paper it should be explained also in the text.
Other minor considerations:
- References 10 on line 44 and 11 on line 45 are wrong
- Figure 3 is presented in the text (line 169) before Figure 2
- In the explanation of Table 1, (lines 256-265) some numbers are given with two decimal places and others with only one. You should follow consistent criteria
- In line 293 when it says “the bias and STDV slightly decrease to -0.008°C and 0.302°C” please check because it seems that it should say “increase”
Author Response
Reviewer 1
This paper investigates the relationship between Hot Events occurrence and diurnal SST amplitude (dSST) variation in the Western Equatorial Pacific by using satellite data. In order to do this the authors also present the methodology used to produce dSST data from satellite sensors and their validation. The research is interesting and well founded, but the article suffers from a lack of clarity, especially in the presentation of the results and their discussion.
It is obvious from reading the work that the authors start from a significant volume of information and data and that they have based their results on solid previous works. However, it is sometimes difficult to distinguish between the original results presented in this paper and previously published results, in part this is because the reader is frequently referred to figures from other papers (references 18 and 20) that are not shown in the document.
R: Thank you for your time and effort in reviewing our paper, and for providing the valuable suggestions, comments, and corrections, which helped make our manuscript stronger. We have modified the manuscript based on the reviewer’s suggestions and hope that the revision adequately addressed the reviewer’s concerns, so that the revised manuscript will be suitable for publication. The modified parts in the manuscript are highlighted in yellow.
We recommend a REWRITE of the text taking into account the following recommendations:
- The objectives of the paper should be more clearly and extensively explained in the Introduction. It would be useful to explain whether the period of study is (2003-2011) or when and why the authors extend the period (1991-2002)
R: Thank you very much for your correction. We apologize for this confusion. Our period of observation is 2003-2011. We have deleted the extended priod (1991-2002).
- In section 2 (Dataset and Methods) we suggest splitting the subsection “2.1.2. Intercomparison data from in situ and geostationary satellite observations” into two subsections, one to describe better the need for using in situ data and their characteristics or limitations, and the other to describe the satellite observations.
R: Thank you very much for your suggestion. We have splitted the subsection 2.1.2. into 2 parts : Insitu SST data (L164) and Geostationary satellite-based SST data (L173).
- We recommend dividing Section 3 Results and Discussion into two sections (3. Results and 4. Discussion). This would make clearer for the reader which are the new results presented by the authors and then separately compare them with other or previous works.
R: Thank you very much for your suggestion. We have separated the discussion part. The new result of this study and the comparison to the previous studies become easier to distinguish.
- The authors should not be making references to figures from other papers which are not presented in the text. This is done repeatedly: Lines 198-204, Line 308, Lines 331-332, Lines 361.This is very confusing, especially when they make statements such as: Line 309-310 “the relation between dSST and HE area is more consistent than the result shown by Qin et al. [18] in Fig. 4, i.e. the area of dSST more than 0.5°C was wider than the HE area”.
R: Thank you very much for your suggestion. We have removed mentioning figures from other papers and elaborate what is left from references 18 and 20. The comparison of the present study with the other papers also has been placed in the discussion part to avoid confusion (L360-387).
- In Figure 5, It would be useful to explain how the authors have depicted the area of HE (red line). It seems that the method is explained in a previous study, Wirasatriya et al., but if the Figure shows your results in this paper it should be explained also in the text.
R: Thank you very much for your suggestion. We have added the explanation of the HE area (L309-312).
Other minor considerations:
- References 10 on line 44 and 11 on line 45 are wrong
R: Thank you very much for your correction. We have fixed the reference list.
- Figure 3 is presented in the text (line 169) before Figure 2
R: Thank you very much for your correction. We have changed Fig. 2 and Fig. 3 positions.
- In the explanation of Table 1, (lines 256-265) some numbers are given with two decimal places and others with only one. You should follow consistent criteria
R: Thank you very much for your suggestion. We have made 2 decimals. (L266).
- In line 293 when it says “the bias and STDV slightly decrease to -0.008°C and 0.302°C” please check because it seems that it should say “increase”.
R : Thank you very much for your comment. What we mean by “decrease” refers to error STDV. So we have deleted bias (L297-2

Reviewer 2 Report
This manuscript examines the variation in diurnal sea surface temperature in the West Equatorial Pacific using mathematical calculations and validate using insitu buoy datasets. I think the MS can be accepted after revision as below:
- Introduction, lines 38, the authors state the generation of diurnal sea surface temperature is caused by change of solar heating as a result of earth’s rotation. Isn’t it more simply to say the generation of diurnal sea surface temperature is caused by change of solar heating as a result of day and night differences?
- Line 39 - The range of delta SST can up to 3oC. Where is the location in the world and what conditions can generate such big delta? This should be elaborated.
- Hot events, is this only occur at tropical region? How is the interaction of the oceanographic currents and occurrence of hot events? Is it only occur in summer? For example, the coastal of China is affected by cold currents in winter, does this means hot events can happen? I think the concept of hot events need extensively elaborated. It is far too simple here.
- Line 80 – I don’t think the author is assessing global diurnal SST. They only cover tropical western Pacific region?
- Line 127, what is SSW? I think the author should type SSW is Sea surface wind?
- 1. Daily mean SR – daily mean solar radiation should be added in figure legend. What is the numerical contours? Need explain. How to derive this figure?
- The authors use Buoy data, which cover numerous buoy data in different tropical region of global waters and use this to validate the delta SST they derive. I think the author should separate the buoys in Atlantic Ocean, Pacific Ocean and Indian Ocean and validate their dataset in these 3 major ocean separately. Also, what is the months the buoy dataset is used? Only summer or only winter? Such validation should separate winter and summer I suppose.
- The following paper also mention there are deviation of satellite derived sea surface temperature from actual measurement. The author should cite this in introduction or discussion.
Meneghesso C., R. Seabra, B. R. Broitman, D. S. Wethey, M. T. Burrows, B. K. K. Chan, T. Guy-Haim, P. A. Ribeiro, G. Rilov, A. M. Santos, L. L. Sousa and F. P. Lima (2020). Remotely-sensed L4 SST underestimates the thermal fingerprint of coastal upwelling. Remote Sensing of Environment, 237 111588.
- The sea surface temperature of coastal waters can also affected by weather and tidal level. So, the delta SST is using in coastal region may not be as the equation derived from the present MS. The authors should also cite the below and state that coastal SST can have greater variation than open ocean SST in introduction/ discussion. The paper below, when the thermal sensor cover by water during high tides, this is the measurement of SST, which indicate even at different tidal levels, the SST can has great variation.
Wang H.-Y., L. M. Tsang, F. P. Lima, R. Seabra, M. Ganmanee, G. A. Williams and B. K. K. Chan (2020). Spatial variation in thermal stress experienced by barnacles on rocky shores: the interplay between geographic variation, tidal cycles and microhabitat temperatures. Frontiers in Marine Science, 7. https://doi.org/10.3389/fmars.2020.00553
- The buoy dataset is huge, the author should state in detail how to access this dataset? Is it downloable from the web? Or should the author put this data in supplementary files for readers to see how the dataset looks like.
- Fig 5. How to plot these graphs, what dataset the authors is using? The solar radiation and wind speed, how to asses such fine spatial scale data?
- Fig 6. The contour is super un-clear, not acceptable. What is 5 means relating to possibility? 5%? The countour need to redraw with clearer colour.
- Discussion/Conclusion – the West Pacific – has many complicated currents, Kuroshio, equatorial Currents, Coastal currents, South China Sea Warm current. The authors has not discussed the delta SST and relating to these currents. This is a must for adding in the discussion and add in the conclusion as well.
Author Response
Reviewer 2
This manuscript examines the variation in diurnal sea surface temperature in the West Equatorial Pacific using mathematical calculations and validate using insitu buoy datasets. I think the MS can be accepted after revision as below:
R: Thank you for your time and effort in reviewing our paper, and for providing the valuable suggestions, comments, and corrections, which helped make our manuscript stronger. We have modified the manuscript based on the reviewer’s suggestions and hope that the revision adequately addressed the reviewer’s concerns, so that the revised manuscript will be suitable for publication. The modified parts in the manuscript are highlighted in yellow.
- Introduction, lines 38, the authors state the generation of diurnal sea surface temperature is caused by change of solar heating as a result of earth’s rotation. Isn’t it more simply to say the generation of diurnal sea surface temperature is caused by change of solar heating as a result of day and night differences?
R : Thank you very much for your suggestion. We have changed the sentence following your suggestion. (L39)
- Line 39 - The range of delta SST can up to 3oC. Where is the location in the world and what conditions can generate such big delta? This should be elaborated.
R: Under the extremely high solar radiation and the absence of wind, dSST can be more than 3°C. For example in the Sargasso Sea during summer [1,2], Japan Sea side of the Tsugaru Strait [4].
- Hot events, is this only occur at tropical region? How is the interaction of the oceanographic currents and occurrence of hot events? Is it only occur in summer? For example, the coastal of China is affected by cold currents in winter, does this means hot events can happen? I think the concept of hot events need extensively elaborated. It is far too simple here.
R : Thank you very much for your question. Hot Event is categorized as an extremely high SST phenomenon in the tropical region. The series of HE studies done by Qin et al ([18]; 2008; 2009a,b; 2010) and Wirasatriya et al. [20,21,43] have shown that the generation of HE is purely driven by air-sea heat exchange which is dominantly by short wave radiation and latent heat flux. Thus, only the condition of high solar radiation and low wind speed could generate HE in the tropical region (L70). In the tropical region, seasons affect the area of HE occurrence. As shown in Fig. 6, the area with the frequent HE occurrence shift northward (southward) from equator during summer (winter) (L327-334). Thus, in the Coastal China during winter is not favorable for HE generation.
- Line 80 – I don’t think the author is assessing global diurnal SST. They only cover tropical western Pacific region?
R: Thank you very much for your question. Here is our clarification. We produced dSST globally and validated with global mooring buoy data. Then we used it for investigating HE which only in the western equatorial Pacific . We have clarified in the abstract (L24-25)
- Line 127, what is SSW? I think the author should type SSW is Sea surface wind?
R: Yes, it is Sea Surface Wind. This has been mentioned in L102.
- 1. Daily mean SR – daily mean solar radiation should be added in figure legend. What is the numerical contours? Need explain. How to derive this figure?
R: Thank you very much for your correction. We have fixed the legend which includes explaining numerical contours.
- The authors use Buoy data, which cover numerous buoy data in different tropical region of global waters and use this to validate the delta SST they derive. I think the author should separate the buoys in Atlantic Ocean, Pacific Ocean and Indian Ocean and validate their dataset in these 3 major ocean separately. Also, what is the months the buoy dataset is used? Only summer or only winter? Such validation should separate winter and summer I suppose.
R: Thank you very much for your question. Since the area for dSST production is a global ocean, we used all buoy in the Atlantic, Indian and Pacific Oceans for validation for both seasons, summer and winter. Specifically for this HE study, we refocus and revalidate the dataset with the buoys in the western equatorial Pacific (L296-300).
- The following paper also mention there are deviation of satellite derived sea surface temperature from actual measurement. The author should cite this in introduction or discussion.
Meneghesso C., R. Seabra, B. R. Broitman, D. S. Wethey, M. T. Burrows, B. K. K. Chan, T. Guy-Haim, P. A. Ribeiro, G. Rilov, A. M. Santos, L. L. Sousa and F. P. Lima (2020). Remotely-sensed L4 SST underestimates the thermal fingerprint of coastal upwelling. Remote Sensing of Environment, 237 111588.
R: Thank you very much for your suggestion. We have added this reference to the discussion (L359).
- The sea surface temperature of coastal waters can also affected by weather and tidal level. So, the delta SST is using in coastal region may not be as the equation derived from the present MS. The authors should also cite the below and state that coastal SST can have greater variation than open ocean SST in introduction/ discussion. The paper below, when the thermal sensor cover by water during high tides, this is the measurement of SST, which indicate even at different tidal levels, the SST can has great variation.
Wang H.-Y., L. M. Tsang, F. P. Lima, R. Seabra, M. Ganmanee, G. A. Williams and B. K. K. Chan (2020). Spatial variation in thermal stress experienced by barnacles on rocky shores: the interplay between geographic variation, tidal cycles and microhabitat temperatures. Frontiers in Marine Science, 7. https://doi.org/10.3389/fmars.2020.00553
R: Thank you very much for pointing out this reference. We have added this reference to the discussion (L357).
- The buoy dataset is huge, the author should state in detail how to access this dataset? Is it downloable from the web? Or should the author put this data in supplementary files for readers to see how the dataset looks like.
R: Thank you very much for your question. The detailed explanation about buoy data is described in L165-172. Yes, this is publically available and downloadable data. The link how to access the data is provided in the acknowledgment. (L441)
- Fig 5. How to plot these graphs, what dataset the authors is using? The solar radiation and wind speed, how to asses such fine spatial scale data?
R: Thank you very much for your question. We used six-hourly reanalysis data from the Japanese 25-year Reanalysis (JRA-25)/Japan Meteorological Agency (JMA) Climate Data Assimilation System (JCDAS) on a 1.25° horizontal grid for wind speed [41] and daily net surface solar radiation on a 1° × 1° grid for 2003–2009 from the International Satellite Cloud Climatology Project (ISCCP) dataset [42]. The grid intervals of these datasets were interpolated into 0.25° to match with NGSST-O data (L196-200). The links for dowloading these datasets are provided in the ackowledgment (L444-445).
- Fig 6. The contour is super un-clear, not acceptable. What is 5 means relating to possibility? 5%? The countour need to redraw with clearer colour.
R: Thank you very much for your suggestion. We have revised this figure. The 5% in a grid means the percentage of HE occurrence during 2003-2011.
- Discussion/Conclusion – the West Pacific – has many complicated currents, Kuroshio, equatorial Currents, Coastal currents, South China Sea Warm current. The authors has not discussed the delta SST and relating to these currents. This is a must for adding in the discussion and add in the conclusion as well.
R: Thank you very much. As has been reviewed by Kawai and Wada [8], the diurnal variation of SST is only influenced by air-sea heat exchange especially related to the variation of solar radiation and wind speed. Thus, this phenomenon is not related with the ocean current system. The high solar radiation and low wind speed lead to the high dSST. This makes high dSST corresponds to the occurrence of HE in western equatorial Pacific as investigated in this study.
Additional references
Qin, H., H. Kawamura, F. Sakaida, and K. Ando (2008), A case study of the tropical hot event in November 2006 (HE0611) using a geostationary meteorological satellite and the TAO/TRITON mooring array, J. Geophys. Res. 113, C08045, doi:10.1029/2007JC004640.
Qin, H., and H. Kawamura (2009a), Surface heat fluxes during hot events, J. Oceanogr. 65, 605-613.
Qin, H., and H. Kawamura (2009b), Atmosphere response to a hot SST event in November 2006 as observed by AIRS instrument, Adv. Space. Res. 44, 395-400, doi:10.1016/j.asr.2009.03.003.
Qin, H., and H. Kawamura (2010), Air-sea interaction throughout the troposphere over a very high sea surface temperature, Geophys. Res. Let., 37, 1-4. doi:10.1029/2009GL041685.
Round 2
Reviewer 1 Report
The comments and suggestions were adressed and the paper can be accepted
Reviewer 2 Report
The comments were addressed and the MS can be accepted.